# Towards Whole Health Toxicology: In-Silico Prediction of Diseases Sensitive to Multi-Chemical Exposures

**DOI:** 10.3390/toxics10120764

**Published:** 2022-12-08

**Authors:** Olatunbosun Arowolo, Victoria Salemme, Alexander Suvorov

**Affiliations:** 1Department of Environmental Health Sciences, School of Public Health and Health Sciences, University of Massachusetts, 686 North Pleasant Street, Amherst, MA 01003, USA; 2Department of Pharmacology, University of California, 1275 Med Science, Davis, CA 95616, USA

**Keywords:** syndromes, xenobiotics, in-silico, multi-chemical, mixed exposures

## Abstract

Chemical exposures from diverse sources merge on a limited number of molecular pathways described as toxicity pathways. Changes in the same set of molecular pathways in different cell and tissue types may generate seemingly unrelated health conditions. Today, no approaches are available to predict in an unbiased way sensitivities of different disease states and their combinations to multi-chemical exposures across the exposome. We propose an inductive in-silico workflow where sensitivities of genes to chemical exposures are identified based on the overlap of existing genomic datasets, and data on sensitivities of individual genes is further used to sequentially derive predictions on sensitivities of molecular pathways, disease states, and groups of disease states (syndromes). Our analysis predicts that conditions representing the most significant public health problems are among the most sensitive to cumulative chemical exposures. These conditions include six leading types of cancer in the world (prostatic, breast, stomach, lung, colorectal neoplasms, and hepatocellular carcinoma), obesity, type 2 diabetes, non-alcoholic fatty liver disease, autistic disorder, Alzheimer’s disease, hypertension, heart failure, brain and myocardial ischemia, and myocardial infarction. Overall, our predictions suggest that environmental risk factors may be underestimated for the most significant public health problems.

## 1. Introduction

Traditional approach to healthcare is based on a reductionist ideology that divides a person into smaller components that are then used individually as subjects of diagnostics and therapeutic interventions [1,2]. This approach contradicts the modern understanding of the integrated nature of biological systems and health. The concept of whole-person health was suggested recently to overcome the drawbacks of prevailing reductionism [3]. This concept suggests that global health problems are not independent of each other and calls for a multidimensional and integrated approach to care to encompass internal (body) and external factors that play a role in the general well-being of an individual [1,3].

The whole person health concept builds a fruitful framework for toxicological research (Figure 1A). Indeed, humans are exposed to hundreds of environmental chemicals daily [4,5]. In one day, exposure may include pesticides, preservatives, and persistent organic pollutants from contaminated foods, stabilizers and solvents from care products and cosmetics, inhaled toxic products of combustion of organic matter from car exhaust gases, flame retardants from furniture, plasticizers from household polymers, a broad range of additives in prescription drugs and other [6,7,8,9]. Over the years, many research studies have demonstrated the impacts of exposures to individual environmental chemicals on human health [5,10,11]. However, these studies do not provide sufficient knowledge to predict the effects of multi-chemical exposures on human health [5,12]. A combination of two or more environmental chemicals may have a significant impact on health either through synergistic, potentiating, or antagonistic effects [5]. For instance, cumulative exposure to 15 pesticides and phthalates, when each of these chemicals was present at no observable adverse effect levels (NOAEL) caused an adverse effect on the male reproductive development in pups of exposed rats [13].

Chemical exposures from diverse and numerous sources merge on a limited number of molecular pathways [4], which in recent years were described as toxicity pathways or adverse outcome pathways (AOP) [14,15,16,17,18]. Changes in the same set of molecular pathways in different cell and tissue types may generate seemingly unrelated health conditions. For example, activation of the aryl hydrocarbon receptor (AhR) pathway may contribute to the development of skin lesions (chloracne) [19], atherosclerosis [20], chronic kidney disease [21], tumor promotion [22], pathological immune response [23], disruption of spermatogenesis [24] and others. Today, the healthcare system will likely treat all these conditions as completely unrelated even if they all are found in the same patient; while recognition of these conditions as a syndrome may facilitate the identification of a causal route of the health problems to inform the most efficient therapeutic interventions.

The number of patients suffering from multimorbidity is increasing globally [2,25,26,27,28]. Some studies developed complex networks between multiple human diseases but with no relation to environmental chemical exposures [26,29,30,31,32,33]. The approach to understanding the complex interactions between different diseases caused by exposure to multiple environmental chemicals is mostly unknown today. The development of this approach is complicated by the fact, that syndromes resulting from the disruption of individual molecular pathways are not well characterized and by a lack of approaches that may integrate information on the disruption of multiple pathways.

In this study, we propose an in-silico approach (Figure 1B) to predict disease states caused by exposures to multiple chemicals using chemical-gene interaction data from the Comparative Toxicogenomic Database (CTD) [34]. 

## 2. Results

The results of the identification of genes’ and pathways’ sensitivity to chemical exposures were reported in our previous study [4]. Therefore, here we focus on disease-level findings. In total 86 KEGG pathways and 194 Reactome pathways matched our selection criteria (FDR q > 0.01 and NES > 1.9) for the identification of associated diseases. After filtering, 1850 diseases were associated with KEGG pathways, and 2481 diseases were associated with Reactome pathways. Following dimension reduction, 21 non-neoplastic diseases and 19 neoplastic diseases associated with chemically sensitive Reactome pathways, as well as 51 non-neoplastic diseases associated with chemically sensitive KEGG pathways were selected for further analysis, see Mendeley data [35].

### 2.1. Disease Categories Associated with Chemically Sensitive Pathways

The lists of the top-ranking diseases based on both the number of inferred pathways (Figure 2 A,C) and the number of inference genes (Figure 2B,D) were highly similar and/or overlapping for both KEGG (Figure 2A,B) and Reactome (Figure 2C,D) curations. In all four analyses, the lists of the top-ranking diseases were dominated by various neoplasms including breast, stomach, prostatic, lung, colorectal neoplasms, hepatocellular carcinoma, and others. Among non-neoplastic conditions, the top-ranking diseases included metabolic disorders (obesity, type 2 diabetes, non-alcoholic fatty liver disease), neurodevelopmental and psychiatric disorders (autistic disorder, schizophrenia, Alzheimer’s disease, hyperalgesia), cardiovascular (hypertension, heart failure, cardiomyopathies, cardiomegaly, brain and myocardial ischemia, myocardial infarction, and reperfusion injury), and immune pathology (contact dermatitis, arthritis juvenile and rheumatoid and HIV infection). Among these, type 2 diabetes, hypertension, and autistic disorder were among the top conditions in all 4 types of analysis (Figure 2).

### 2.2. Clusters of Disease Categories Sharing Similar Chemically Sensitive Pathways

For this analysis, only these diseases were taken which were strongly associated with big numbers of chemically sensitive pathways. In other words, these are conditions that are most likely to be induced or exacerbated by cumulative chemical exposures. The results of Self Organizing Map (SOM) and hierarchical clustering are shown in Figure 3 for diseases clustering together based on the similarity of overlapping chemically sensitive Reactome pathways. The same results for non-neoplastic disease grouping based on KEGG pathways are shown in the Appendix A. 

As illustrated in Figure 3A,B, in both SOM and hierarchical clustering the mechanistic similarity of many non-neoplastic diseases clustering together is not obvious at a first glance. However deeper analysis of existing literature reveals common molecular etiology or other associations between many conditions that cluster together. For example, Type 2 Diabetes and hypertension are common comorbidities and they share many common mechanisms including upregulation of the renin-angiotensin-aldosterone system, oxidative stress, and inflammation [36]. Autism, rheumatoid arthritis, and inflammation are all connected via the pathological response of the immune system. Maternal inflammation is a significant risk factor for autistic disorder in offspring [37,38] and rheumatoid arthritis in parents is positively associated with the risk of autistic disorder in offspring [39]. Obesity is associated with excessive accumulation of lipids in the liver (non-alcoholic fatty liver disease—NAFLD) and NAFLD is a leading cause of liver cirrhosis, where the latter is the terminal stage of disease progression [40]. Links between some other diseases are more obvious. Some examples of such groups of diseases associated with chemically sensitive KEGG pathways include RASopathies: Noonan Syndrome, Costello Syndrome, and Cardiofaciocutaneous Syndrome; diseases of lungs: asthma and pneumonia; conditions associated with the use of drugs of abuse: cocaine-related disorders and substance withdrawal syndrome, diseases associated with fibrotic scarring: fibrosis, pulmonary fibrosis and NAFLD and others (Appendix A). The links between some other conditions that are stably grouped across different comparisons are not obvious. One such example is the clustering together of obesity and liver conditions (NAFLD, cirrhosis) with endometriosis and spontaneous abortion. 

Hierarchical clustering of neoplastic diseases is more difficult to interpret as different forms of cancer are associated with perturbations of a similar set of mechanisms, ‘cancer hallmarks’ [41,42,43] while differences between molecular pathways affected in different forms of cancer may be subtle. Importantly, close to synonymic terms—neoplasm metastasis and neoplasm invasiveness—clustered together in both SOM and hierarchical clustering. Additionally prostatic and breast neoplasms clustered together. Both conditions are known to be sex steroid-dependent cancers with the same genetic and environmental risk factors [44].

## 3. Discussion

The current study is an attempt to identify in an unbiased way individual diseases and disease groups highly sensitive to cumulative chemical exposures. The approach used in this study is based on the in-silico inductive workflow where first genes sensitive to chemical exposures are identified based on the overlap of existing genomic studies. In the next step, molecular pathways enriched with highly sensitive genes are identified. Then sensitivity of diseases to chemical exposures is quantified based on their overlap with sensitive pathways. And finally, sensitive diseases are grouped based on the similarity of overlapping sensitive pathways. The initial steps of this analysis including the identification of the sensitive genes and pathways were published in a separate study [4]. The current study reports the results of the prediction of diseases and disease clusters sensitive to cumulative chemical exposures. Using our unbiased approach, we predict that conditions most sensitive to cumulative chemical exposures include different neoplasms, metabolic disorders, neurodevelopmental and psychiatric disorders, and cardiovascular and immune pathology. 

Our analysis suggests that leading neoplasms with environmental etiology include prostatic, breast, stomach, lung, colorectal neoplasms, and hepatocellular carcinoma. The importance of these findings is determined by the fact that these neoplasms are the six leading types of cancer with the highest incidence worldwide [45]. The environmental etiology of these forms of cancer is supported by extensive previous research,—see for example recent reviews [46,47,48,49,50,51]. Among non-neoplastic conditions, the top-ranking predicted diseases represent the major public health problems, such as obesity, type 2 diabetes, non-alcoholic fatty liver disease, autistic disorder, Alzheimer’s disease, hypertension, heart failure, brain and myocardial ischemia, and myocardial infarction. Along with cancer, four of these conditions (heart disease, stroke, Alzheimer’s disease, and diabetes) are the major causes of death in the USA [52] and the world [53]. 

Other conditions have high importance for the public health system due to their prevalence and upward trends in their incidence. For example, over 1 billion people are obese today [54,55] and if this trajectory continues, 38% of the world’s adult population will be overweight and another 20% will be obese by 2030 [56]. Although, association between exposures to some individual xenobiotics and obesity was shown before [57], our study is first to predict that obesity is sensitive outcome to cumulative exposure across the exposome. Around 462 million individuals are suffering from type-2 diabetes, accounting for 6.3% of the world’s population [58]. NAFLD is the most common liver disease in the world [59], with about 24% of Americans diagnosed with NAFLD [60]. The Center for Diseases and Prevention (CDC) reported that 1 in every 44 eight years old children are diagnosed with autistic disorder in 2018 [61]. 

Overall, our predictions suggest that environmental risk factors may be underestimated for the most significant public health problems. These findings provide grounds for optimism, assuming that significant progress may be achieved in mitigation of the major public health problems, including major causes of death in the world, by merely better management of chemical exposures. 

Chemical exposures from diverse sources merge their effects on a limited number of molecular pathways [4]. Changes in the same set of molecular pathways in different cell and tissue types may generate seemingly unrelated health conditions [62]. Today, the healthcare system will likely treat these conditions as unrelated, focusing mostly on the symptoms of each condition rather than the common root causes [2,63]. This approach does not match modern global reality, where both multiple exposures [64] and multimorbidity [65,66] are becoming a new norm. For example, 81% of Americans ≥ 65 years and 50% of those between 45-and 65 years were reported to have multiple chronic conditions [67]. It was also projected that by 2035, 17% of the UK population would suffer from four or more chronic conditions [68]. 

Thus, for toxicology, there is an urgent need to identify approaches to predict clusters of health conditions that can be induced by multi-chemical exposures. To our knowledge, this paper is the first attempt to predict syndromes generated from exposure to multiple chemicals in an unbiased manner. This approach may provide a useful framework for the whole person health concept which can inspire translational research driven toward better health care interventions. 

Analysis of effects of mixed exposures in human population studies is a significant challenge due to the infinite number of mixtures compositions and their heterogenicity in terms of biological activities of individual components. The widely used approach to study effects of mixed exposures consists in the reduction of the number of individual compounds based on the similarity of their mode of action (MOA), where cumulative toxicity of xenobiotics with a similar MOA in a mixture is expressed as a single number [69,70]. This approach is limited however only to these compounds for which molecular mechanisms of toxicity are well understood and are highly similar. Our study suggests that an alternative approach for the analysis of health effects of mixed exposures in human populations may focuse on these health outcomes, which are known to be the most sensitive to cumulative chemical exposures regardless of the composition of mixtures. Feasibility of this approach is supported by our observation, that identification of sensitive genes and pathways using differtent sets of chemical-gene interaction datasets with non-overlapping lists of chemical compounds produce almost same results [4]. In other words, our data suggest, that as soon as there is a big number of chemicals in the mixture, the same genes, pathways and health outcomes will be affected regardless of the mixture composition.

Our study predicted clusters of health conditions that may result from perturbations of similar molecular pathways sensitive to chemical exposures. Even though in most instances our approach provides information concordant with existing scientific findings, it is difficult to evaluate the quality of our prediction as no independent method is available. For instance, our model predictably identified clusters of diseases with known shared pathology (e.g., asthma and pneumonia), but also it identified some clusters which are difficult to explain (e.g., liver cirrhosis and cleft palate). In addition, at the step of identification of chemoically-sensitive genes we did not consider doses of chemicals used in the original studies [4]. We assume that the value of our effort to identify syndromes induced by chemical exposures will be tested in the future when other in-silico, in-vitro, in-vivo, and epidemiological/clinical studies will be done to target the same research question.

## 4. Materials and Methods

### 4.1. Identification of Genes Sensitive to Chemical Exposures

Our previous studies developed an unbiased approach to identify genes sensitive to chemical exposures [4,71,72]. In short, data on chemical-gene interaction (CGI) was collected from the CTD using the following criteria. Firstly, gene expression data were extracted from toxicological experiments in which transcriptomic responses to chemical compounds were analyzed using high-throughput techniques. In total data from 2169 individual sources were extracted, covering experiments with 1239 chemical compounds. We selected only in vivo and in-vitro studies that used human, rat, or mouse cells or tissue for their gene expression analysis. All genes that were not present in the genomes of all three species (human, rat, and mouse) were excluded from the analysis. This extraction criterion resulted in a database of 591,084 CGI. The number of CGI was calculated for each of 17,338 genes in the database to represent its sensitivity to chemical exposures. It is important to note that ranked sensitivities of genes to chemical exposures do not depend on the composition of chemicals used in original studies—sources of transcriptomic data [4]. The full list of genes with their corresponding CGI numbers is available through Mendeley Data [35].

### 4.2. Identification of Molecular Pathways Enriched with Chemically Sensitive Genes

Gene Set Enrichment Analysis (GSEA) was used to identify molecular pathways sensitive to chemical exposures. For this analysis, the list of genes with their corresponding CGI numbers was analyzed against two independent open-access databases for pathway curation: KEGG [73] and Reactome [74,75]. The details of this analysis are described elsewhere [72]. GSEA is an efficient method for the identification of changes in biological pathways as it captures cumulative changes of multiple members of a pathway [76]. The details of the methods and statistical approaches used by GSEA are described elsewhere [76,77,78,79]. The following stringent criteria were used to identify significant molecular pathways sensitive to chemical exposures: false discovery rate (FDR) q > 0.01 and normalized enrichment score (NES) > 1.9. The results of the GSEA analysis are available through Mendeley Data [35].

### 4.3. Identification of Disease States Sensitive to Chemical Exposures

To identify disease categories that are sensitive to chemical exposures, the lists of significantly enriched KEGG and Reactome pathways were submitted to the CTD to run pathway-disease association analysis. This analysis retrieves disease associations inferred based on shared genes among CTD-curated gene-disease associations and gene-pathway associations established by KEGG and Reactome curation [34]. This step resulted in a matrix of shared gene numbers between chemically sensitive pathways and disease states. All disease states identified from experimental or animal models (e.g., experimental liver cirrhosis, animal muscular dystrophy, experimental melanoma), as well as very general disease categories lacking specificity (e.g., neoplasms, wounds and injuries, drug-related side effects, and adverse reactions) and non-human diseases (Q-fever), were excluded from further analysis. The filtered pathway-disease matrices for KEGG and Reactome pathways are shown in Mendeley data [35]. 

Two numeric values were used to quantify the sensitivity of disease states to chemical exposures (Figure 4): the number of inferred pathways associated with the disease state and the sum of genes from every pathway overlapping with the disease (number of inference genes). One disadvantage of this second metric is that some genes, members of multiple pathways, may be counted more than one time. The lists of disease categories with their corresponding numbers of inferred pathways and inference genes are shown in Mendeley data [35].

### 4.4. Dimension Reduction of the Pathway-Disease Matrix

The disease-pathway matrices for KEGG and Reactome pathways were split into neoplastic and non-neoplastic diseases. For each of the resulting matrices, we then identified the top sensitive diseases that have the highest numbers of inference genes, using a method for the identification of cut-off points in descriptive high-throughput omics studies [80]. This approach assumes that only a small number of diseases with high numbers of inference genes in the disease-pathways interactions are the dominating conditions induced/exacerbated by multi-chemical exposures in the general population. The ranked distribution of the inference gene numbers for the KEGG neoplastic diseases matrix was not biphasic (exponential/super-exponential), a requirement of the method for a cutoff point identification. Therefore, this dataset was excluded from further analysis. 

### 4.5. Prediction of Syndromes Induced by Multi-Chemical Exposures

Following dimension reduction, diseases with the highest overlap with chemically sensitive pathways were used to predict disease groups (syndromes) that may be induced/exacerbated due to the perturbation of similar sets of molecular pathways by multi-chemical exposures. The self-organizing map (SOM) approach was used to map each disease category into a 2D plot [81]. SOM is a powerful statistical tool used for visualizing high-dimensional data, and exploring and clustering samples while preserving their distances [81,82,83]. Since the map is self-organized, the method determines the maximum number of dimensions that may be used based on the data. We further reduced dimensions to minimize the numbers of empty cells/nodes [84,85,86]. In this analysis diseases associated with similar molecular pathways are clustered into a single cell. The SOM distant neighbor plot (U-matrix) was used to identify the relatedness of each cell to another cell. To validate our predictions using an alternative approach, hierarchical clustering of the pathway-disease matrices was also done to identify groups of diseases associated with similar chemically sensitive pathways. All analyses were done in RStudio (2022.02.3 + 492 “Prairie Trillium”) using the Kohonen package for SOM, factoextra, and cluster package for hierarchical clustering (codes are provided in the Appendix A). 

## 5. Conclusions

In this study, we developed an unbiased approach to predict diseases sensitive to multi-chemical exposures and clusters of such diseases (syndromes) resulting from chemical perturbation of the same sets of molecular pathways. Our analysis indicates that diseases representing major public health problems today may be among the most sensitive to chemical exposures. Today it is difficult to test the biological relevance of syndromes predicted by our analysis as no alternative methods are available. 

## Figures and Tables

**Figure 1 toxics-10-00764-f001:**
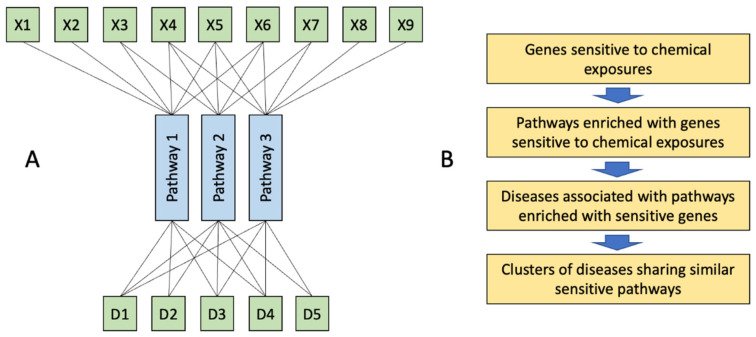
The whole person approach in toxicology. (**A**), Multiple xenobiotics (X) affect limited number of molecular pathways. Disruption of these pathways in different cells across the organisms may result in the development of seemingly unrelated diseases (D) which represent a syndrome of chemical exposure. (**B**), Outline of this study: genes sensitive to chemical exposures, identified by overlaying transcriptomic datasets from toxicological studies were used to identify pathways sensitive to chemical exposures. These pathways were further used to identify associated diseases. Finally, clusters of diseases sharing similar pathways were identified to predict syndromes of chemical exposures.

**Figure 2 toxics-10-00764-f002:**
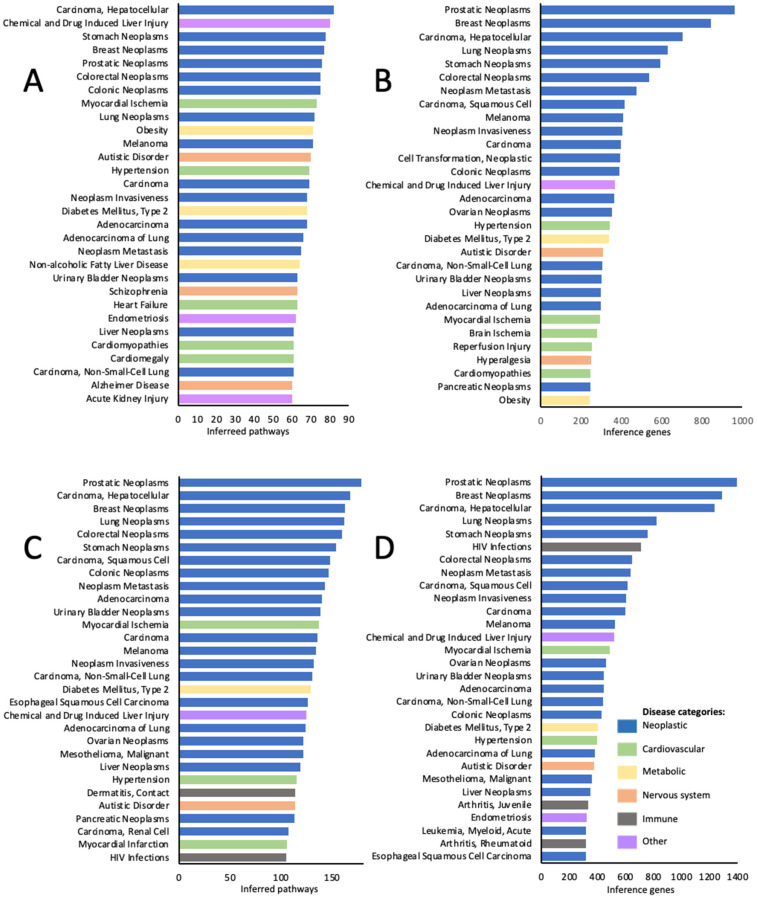
Top 30 disease categories out of 1850 associated with KEGG pathways (**A**,**B**), and out of 2481 associated with Reactome pathways (**C**,**D**) sensitive to chemical exposures. A and C—top ranking diseases identified based on the number of inferred pathways; B and D—top ranking diseases identified based on the number of inference genes.

**Figure 3 toxics-10-00764-f003:**
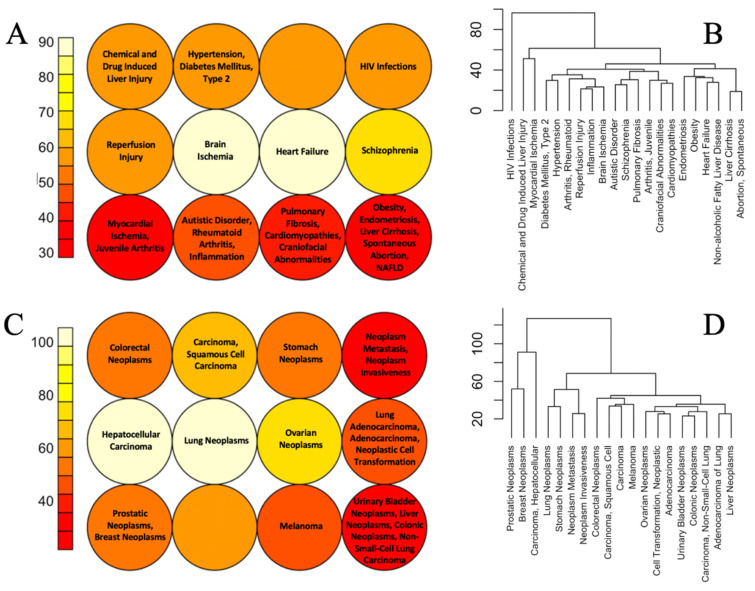
SOM neighbor distant plots (**A**,**C**) and hierarchical clustering (**B**,**D**) of non-neoplastic (**A**,**B**) and neoplastic (**C**,**D**) diseases based on the similarity of their overlap with chemically sensitive Reactome pathways. In SOM plots, cells of similar color contain related diseases.

**Figure 4 toxics-10-00764-f004:**
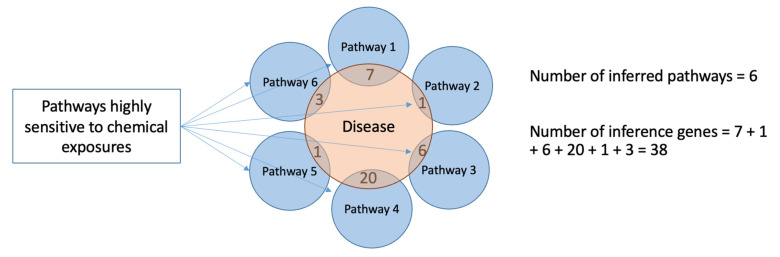
Two approaches for quantification of diseases’ sensitivity to chemical exposures. The first approach is based on the quantification of chemically sensitive pathways associated with the disease state—the number of inferred pathways. The figure illustrates an example, where a disease (testaceous circle) is associated with six inferred pathways (blue circles). The second approach counts the sum of genes from every pathway overlapping with the disease genes (number of inference genes).

## Data Availability

All data used in this study are available through Mendeley Data [35].

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
