# Peer review of "Towards Whole Health Toxicology: In-Silico Prediction of Diseases Sensitive to Multi-Chemical Exposures"

_toxics, 2022, doi:10.3390/toxics10120764_

Round 1
Reviewer 1 Report
This is the revision of the manuscript titled "Towards whole Health Toxicology: In-silico Prediction of Diseases Sensitive to Multi-chemical Exposures" by Ola. Arowolo et al. In general, the concept of the manuscript is interesting. The text is well written and well organised. The work is appropriate for the Journal and the results are interesting. There are only minor observations that must be addressed before the manuscript can be accepted.
Figure 1 should be shifted. Figures should be placed in the article next to relevant text.
L119: the abbreviation must be defined.
Author Response
This is the revision of the manuscript titled "Towards whole Health Toxicology: In-silico Prediction of Diseases Sensitive to Multi-chemical Exposures" by Ola. Arowolo et al. In general, the concept of the manuscript is interesting. The text is well written and well organised. The work is appropriate for the Journal and the results are interesting. There are only minor observations that must be addressed before the manuscript can be accepted.
Thank you for the positive comments on our manuscript
Figure 1 should be shifted. Figures should be placed in the article next to relevant text.
The figure has been shifted as advised
L119: the abbreviation must be defined.
The abbreviation is now defined in L141
Reviewer 2 Report
The manuscript describes a suitable in-silico approach to predict diseases sensitive to multiple chemical exposures. Initially, authors identified genes highly sensitive to chemical exposures and then the molecular pathways enriched with that genes. Finally, they grouped sensitive diseases on the basis of overlapping sensitive pathways.
This is a very interesting approach since population is exposed to multiple chemicals at the same time and each of chemical is able to affect a number of organs linked to different pathologies. Indeed, the evaluation of combined exposure on health by epidemiologic or by in vitro and in vivo studies is to be increased taking into account the additive synergistic or antagonistic effects of chemicals also at low doses.
However, several diseases indicated by authors as results of their study are known to be associated to chemical exposure even considering a single chemical, for example: Kim KY, Lee E, Kim Y. The Association between Bisphenol A Exposure and Obesity in Children-A Systematic Review with Meta-Analysis. Int J Environ Res Public Health. 2019 Jul 15;16(14):2521. doi: 10.3390/ijerph16142521. PMID: 31311074; PMCID: PMC6678763.
In my opinion, besides the approach, the novelty is the identification of common pathways that can be considered a possible biomarkers linking exposure and diseases resulting from this study; this could be an issue to be developed and discussed in the text.
Moreover, what it is missing in the text is precisely a paragraph focused on chemical mixtures. The authors considered data on gene expression derived from studies covering experiments with 1,239 chemicals; therefore, It seems that the considered studies used a single compound. Thus, the question is in what way authors assumed the exposure to a mixture? Probably based on the common target organ, or what criterion have they applied to consider a multiple chemical exposure? Have they considered also the level of exposure? In my opinion, the manuscript should be revised accordingly.
Some items need to be revised:
- multi-chemical or multichemical? Please uniform in the whole text, including the title
- line 167: the number of reference is wrong. The reference 45 is the same cited as number 4, please fix the mistake and renumber the references.
Author Response
The manuscript describes a suitable in-silico approach to predict diseases sensitive to multiple chemical exposures. Initially, authors identified genes highly sensitive to chemical exposures and then the molecular pathways enriched with that genes. Finally, they grouped sensitive diseases on the basis of overlapping sensitive pathways.
This is a very interesting approach since population is exposed to multiple chemicals at the same time and each of chemical is able to affect a number of organs linked to different pathologies. Indeed, the evaluation of combined exposure on health by epidemiologic or by in vitro and in vivo studies is to be increased taking into account the additive synergistic or antagonistic effects of chemicals also at low doses.
Thank you for the positive comments on our manuscript
However, several diseases indicated by authors as results of their study are known to be associated to chemical exposure even considering a single chemical, for example: Kim KY, Lee E, Kim Y. The Association between Bisphenol A Exposure and Obesity in Children-A Systematic Review with Meta-Analysis. Int J Environ Res Public Health. 2019 Jul 15;16(14):2521. doi: 10.3390/ijerph16142521. PMID: 31311074; PMCID: PMC6678763.
Thank you for this comment. We included the reference in our text (L210-212). Overall, the value of our study consists in the prediction of health effects from the whole exposome.
In my opinion, besides the approach, the novelty is the identification of common pathways that can be considered a possible biomarkers linking exposure and diseases resulting from this study; this could be an issue to be developed and discussed in the text.
Thank you for the kind suggestion. The discussion of the common pathways was done in our previous study – see reference #4. Based on this knowledge, this current study moves further to identify those diseases which are sensitive to multi-chemical exposures and predict possible interactions that may exist between the diseases. We disclose that in lines 189-190.
Moreover, what it is missing in the text is precisely a paragraph focused on chemical mixtures.
Thank you for this suggestion. We have added this paragraph in the discussion, - see lines 240-256
The authors considered data on gene expression derived from studies covering experiments with 1,239 chemicals; therefore, It seems that the considered studies used a single compound. Thus, the question is in what way authors assumed the exposure to a mixture? Probably based on the common target organ, or what criterion have they applied to consider a multiple chemical exposure? Have they considered also the level of exposure? In my opinion, the manuscript should be revised accordingly.
The criteria for the extraction of original transcriptomic datasets for our analysis are described in methods (4.1) and in more details this is described in our previous study – reference # 4. In short, transcriptomic datasets were taken from CTD with the following criteria (1) we selected data from high-throughput experiments only; (2) we selected data from 3 organisms only (mice, rats, humans); (3) we selected data where gene expression was analysis to one chemical compound. Prediction of effects of mixed exposures is done in-silico, based on overlaps of responses to single chemical exposures. We did not account for the doses of individual chemicals used in the original studies. Unfortunately doses were not available via CTD – the source of our data. This limitation was disclosed in our previous paper (ref. #4) and we have added this disclosure in the current manuscript as well – see lines 263-265.
Some items need to be revised:
- multi-chemical or multichemical? Please uniform in the whole text, including the title
- line 167: the number of reference is wrong. The reference 45 is the same cited as number 4, please fix the mistake and renumber the references.
“multichemical” have been replaced with “multi-chemical” in the whole text. The issues with referencing have been corrected as well.
Reviewer 3 Report
In this study, Arowolo et al. performed a chemical-gene-pathway-disease analysis using Comparative Toxicogenomic Database. From the public health’s perspective, the significance of this study is limited. However, the results may direct future research studying disease comorbidity and chemical co-exposure. My major concerns, comments, and suggestions are detailed as follows:
l The findings from this study were descriptive. The major flaw is that the “prediction” cannot be validated.
l Is it possible to identify the components of multi-chemical exposures that potentially associated with the diseases? What kinds of chemical co-exposure can increase the risk of developing diseases? Can you predict the potential synergistic or potentiating effects based on data from pathway and disease analysis? These would be of great importance for disease prevention
l Please specify the version of the R packages, and the settings used to run the SOM and clustering. I would suggest to prepare a R markdown document as a supplemental file.
Line 128: should be figures 3A and 3B.
Author Response
In this study, Arowolo et al. performed a chemical-gene-pathway-disease analysis using Comparative Toxicogenomic Database. From the public health’s perspective, the significance of this study is limited. However, the results may direct future research studying disease comorbidity and chemical co-exposure. My major concerns, comments, and suggestions are detailed as follows:
The findings from this study were descriptive. The major flaw is that the “prediction” cannot be validated.
Yes, we agree with the reviewer on the fact that our prediction cannot be validated as of now, since there are no current independent methods that may be used for the validation. We understand this limitation of our study and disclose it in lines 257-263. We believe though, that our findings are concordant with the significant body of literature on health effects of environmental exposures.
Is it possible to identify the components of multi-chemical exposures that potentially associated with the diseases? What kinds of chemical co-exposure can increase the risk of developing diseases? Can you predict the potential synergistic or potentiating effects based on data from pathway and disease analysis? These would be of great importance for disease prevention
We agree with the reviewer that identification of the individual compounds – disruptors of a specific pathway/health outcome is important. Of the original 2,169 studies from which transcriptomic datasets were extracted, each responds to this type of question for an individual compound. In this manuscript we are trying to respond to a different question: what diseases are triggered/exacerbated most often by cumulative exposures to all compounds to which humans are exposed in the general population. Thus, the philosophy of this study is opposite to the questions asked by the reviewer. They are of high importance, and we will address these questions in our future studies.
Please specify the version of the R packages, and the settings used to run the SOM and clustering. I would suggest to prepare a R markdown document as a supplemental file.
The version of the R packages has been included in the text, see lines 349-351. Rmarkdown of the settings used in the running of SOM and clustering have been submitted to supplemental files.
Line 128: should be figures 3A and 3B.
The correction has been made, kindly see L150
Round 2
Reviewer 2 Report
Thank you to the authors for modifying the text according to comments and suggestions. Tha paper is accepted in the present form.
Reviewer 3 Report
The authors appropriately addressed the reviewer's comments. Minor typo needs to be fixed:
Line 251: chemically-sensitive